# The Role of Neutrophilic Granulocytes in Philadelphia Chromosome Negative Myeloproliferative Neoplasms

**DOI:** 10.3390/ijms22179555

**Published:** 2021-09-03

**Authors:** Dominik Kiem, Sandro Wagner, Teresa Magnes, Alexander Egle, Richard Greil, Thomas Melchardt

**Affiliations:** 1Oncologic Center, Department of Internal Medicine III with Haematology, Medical Oncology, Haemostaseology, Infectiology and Rheumatology, Paracelsus Medical University, 5020 Salzburg, Austria; d.kiem@salk.at (D.K.); sa.wagner@salk.at (S.W.); t.magnes@salk.at (T.M.); a.egle@salk.at (A.E.); r.greil@salk.at (R.G.); 2Cancer Cluster Salzburg, 5020 Salzburg, Austria; 3Salzburg Cancer Research Institute-Laboratory for Immunological and Molecular Cancer Research (SCRI-LIMCR), 5020 Salzburg, Austria

**Keywords:** myeloproliferative neoplasms, chronic inflammation, neutrophilic granulocytes

## Abstract

Philadelphia chromosome negative myeloproliferative neoplasms (MPN) are composed of polycythemia vera (PV), essential thrombocytosis (ET), and primary myelofibrosis (PMF). The clinical picture is determined by constitutional symptoms and complications, including arterial and venous thromboembolic or hemorrhagic events. MPNs are characterized by mutations in *JAK2*, *MPL*, or *CALR*, with additional mutations leading to an expansion of myeloid cell lineages and, in PMF, to marrow fibrosis and cytopenias. Chronic inflammation impacting the initiation and expansion of disease in a major way has been described. Neutrophilic granulocytes play a major role in the pathogenesis of thromboembolic events via the secretion of inflammatory markers, as well as via interaction with thrombocytes and the endothelium. In this review, we discuss the molecular biology underlying myeloproliferative neoplasms and point out the central role of leukocytosis and, specifically, neutrophilic granulocytes in this group of disorders.

## 1. Introduction

Myeloproliferative neoplasms (MPN) form a distinct group of hematologic malignancies. According to the current WHO classification, the spectrum of MPNs include: chronic myeloid leukemia (CML); chronic neutrophilic leukemia (CNL); polycythemia vera (PV); primary myelofibrosis (PMF); essential thrombocythemia (ET); chronic eosinophilic leukemia not otherwise specified (CEL, NOS); and otherwise unclassifiable MPNs (MPN-U) [1]. In this review, we will focus strictly on the three most frequent Philadelphia chromosome (Ph) negative disease entities: PV, ET, and PMF, which are mostly associated with mutations in either the Janus kinase 2 (*JAK2*) gene, the myelproliferative leukemia virus oncogene (*MPL*), or the calreticulin (*CALR*) gene [2,3]. These three entities share common features in their biology and pathophysiology and lead to partially overlapping clinical presentations. Since CML arises from a chromosome 9:22 translocation, commonly referred to as the Philadelphia chromosome, which therefore displays a significantly different biology compared to PV, ET, and PMF, we omitted the further discussion of that entity in our review [4].

## 2. Clinical Picture and Course of Disease

MPNs are characterized by clonal proliferation, which will be described in Section 3. Although the diagnosis of the exact MPN subtype is based on different criteria regarding peripheral blood cell counts and bone marrow biopsy, clinical symptoms may be very similar. These can include constitutional symptoms (most prominently fatigue and weight loss), bone pain, itching, mood alterations, and abdominal discomfort or pain, as splenomegaly is often present at initial diagnosis [5]. Aquagenic pruritus and erythromelalgia are typical symptoms associated with PV and may precede disease onset for years [6,7]. MPNs represent a biological continuum, and transformation between entities is possible. *JAK2*-positive PV and ET arguably represent the same disease entity with diagnosis determined by allelic frequency and other genetic factors, which will be described in more depth in Section 3 [8]. Although PMF manifests in previously healthy individuals, the transition from PV or ET to post-PV or -ET myelofibrosis (MF) may eventually occur (10-year risk 4.9–6% in PV and 0.8–4.9% in ET) [9,10]. Additionally, at an advanced stage of disease, the risk of transformation into acute myeloid leukemia (AML) increases [11]. Bleeding and thromboembolic events are common complications that have been associated with leukocytosis; treatment generally aims to reduce the rate of thrombo-hemorrhagic events and clinical symptoms [9,12,13,14]. The risk of transformation into AML does not seem to be lowered through the use of currently available treatment options and depends mainly on the subtype of disease: The 10-year transformation risk is less than 1% in ET, around 3% in PV, and 10–20% in PMF [9,15]. In the following, we will further elucidate the roles of neutrophilic granulocytes in MPNs.

## 3. Molecular Biology of Ph-Negative Myeloproliferative Neoplasms and Predictive/Prognostic Implications

MPNs are known to originate from a single, mutated hematopoietic stem cell (HSC), termed the MPN stem cell [3,16]. A hallmark of MPNs is the activation of the (MPL-)JAK-STAT pathway [17,18]. Apart from thrombopoiesis, the MPL-signaling axis plays a key-role in HSC renewal [19]. Additionally, JAK2 is involved in erythropoiesis downstream of the EPO receptor, as well as in granulopoiesis downstream of granulocyte colony-stimulating factor (G-CSF) receptor (Figure 1). The activating mutation *JAK2V617F* [20,21,22,23] is found in almost all patients with PV (with a *JAK2* exon 12 deletion in a small fraction of patients) [24] and around 60% of patients with ET or PMF [25,26]. In ET and PMF, *MPL* mutations are found in 3–10%: most prominent are the mutations *MPLW515L* and *MPLW515K* [26], while *CALR* exon 9 frameshift mutations are detected in 20–25% of patients [26,27]. The latter indirectly leads to a constitutive activation of MPL [28]. In around 8% of patients, no alteration in the aforementioned genes can be found and are called “triple-negative” patients [8].

It is important to note that *JAK2V617F* mutations with a low allelic fraction can be found in a significant number of healthy subjects, therefore, are a common cause for clonal hematopoiesis of indeterminate potential (CHIP) [29] that may precede clinical disease onset by many years [30]. Arguably, however, it confers an elevated risk for portal and mesenteric vein thrombosis in patients not (yet) fulfilling the diagnostic criteria for any MPN [31,32]. The introduction of a *JAK2V617F* mutation into an HSC resulted in myeloid-lineage biased HSC leading to erythro- and thrombocytosis in a mouse model but was also associated with increased DNA damage and low disease penetration [33]. *JAK2V617F* mutations have, however, also been detected within lymphoid cell lineages, implying that malignant stem cells maintain their ability for lymphoid differentiation [34,35]. Even though the *JAK2V617F* mutation is thought to be a driver of disease in affected patients, commonly used mouse models have not shown clonal expansion at the HSC level [36]. Therefore, additional mutations at the HSC level are necessary before MPNs can develop. This may involve regulators of the JAK-STAT pathway or gene expression via epigenetic modifications or transcription factor mutations [37]. As reviewed by several authors, additional mutations might affect cytokine signaling, splicing machinery, transcription factors, and epigenetic modifiers [38,39,40]. Apparently, germline predisposition might also influence disease penetrance as well as phenotype of MPN. Additionally, it is not only the presence of additional somatic mutations, but also the order in which they are acquired that determines the development of disease [41,42]. The disease phenotype is also in part determined by the allelic frequency. For *JAK2V617F*, the allelic burden seems to be lowest in ET and highest in MF; similarly, for *CALR*, the allelic burden is higher in MF than ET [43,44,45]. Several genetic factors carry prognostic implications: A worse prognosis in PV is associated with mutations in *ASXL1*, *SRSF2*, and *IDH2*; in ET, with mutations in *SH2B3*, *SF3B1*, *U2AF1*, *TP53*, *IDH2*, and *EZH2* [46]. Examples for genes associated with poor prognosis in MF include *ASXL-1*, *SRF2*, *U2AF1-Q157*, *TP53*, or *Ras* [8,35,47].

Molecular alterations on the HSC level are the underlying cause of all subtypes of MPN and can be carried across all steps of granulopoiesis, i.e., in common myeloid progenitors, granulocyte monocyte progenitors, promyelocytes, myelocytes, metamyelocytes, and, finally, mature neutrophils.

## 4. Inflammation and Neutrophilic Granulocytes in MPN

In recent years, several studies pointed out the importance of inflammation and neutrophils in the initiation and promotion of MPN [41,48,49,50,51,52]. This includes the expansion of *JAK2V617F* mutated HSCs driven by secretion of proinflammatory cytokines by stromal cells of the bone marrow [52]. Longhitano et al. proposed a role for the inflammasome in MPN [53]. For several inflammatory markers, a positive impact on MPN stem cells was found. Tumor necrosis factor-alpha (TNF-alpha) has been shown to be elevated as a result of *JAK2V617F* mutation and assisted in the expansion of mutant clones, while having a negative effect on the non-mutant HSC [54]. From a murine model, and subsequently from human bone marrow gathered from patients, IL-33 has been identified to induce colony formation on the HSC level and to lead to prolonged survival of *JAK2V617F*-positive cell lines [55]. In an in vitro study, the expansion of erythroid precursors seemed to be dependent on IL-11 and hepatocyte growth factor (HGF), even in the presence of a *JAK2V617F* mutation [56]. Similarly, inflammatory processes may further support the malignant clones even after shutting down JAK2 signaling; *JAK2V617F*-positive cells have been reported to be protected against JAK2 inhibition by IL-6, CXCL10, and fibroblast growth factor (FGF) [57].

The transcription factor NF-E2 is often found to be overexpressed in patients with MPN [58], independently of *JAK2* mutational status. As a mechanism, an increased binding of RUNX-1 to the NF-E2 promotor has been described in neutrophils [59]. Increased levels of NF-E2 in mouse models led to leuko-and thrombocytosis [60]. NF-E2 is also induced by IL-1beta, mainly found in PMF and ET [53], and is shown to induce MF and fibrotic tissue formation in vitro [61], as well as to be secreted by *JAK2V617F*-positive HSC in animal models [62]. Conversely, NF-E2 induces IL-8 [63], which is often elevated in patients with MPN [64,65]. Apart from altering megakaryopoiesis [65], IL-8 has been described as contributing to stem cell mobilization and to neutrophilia [66], as well as stimulating growth of the erythroid lineage [67]. Additionally, granulocytes in MF have been described as a major source of IL-8 themselves [68]. As summarized by Longhitano et al., elevation of many other cytokines, growth factors, and metalloproteases can be found [53]. Importantly, however, in PMF, prognostic values for IL-8, IL-2R, IL-12, IL-15 [69], and CRP (which is directly related to the *JAK2V617F* allele burden, [70,71]) levels, have been described. Similarly, a correlation between CRP and *JAK2V617F* has been described in PV and ET [72]. Although cytokines may be produced by malignant clones, JAK-STAT activation does also occur in non-malignant cells and causes them to further support the inflammatory process [68]. YKL-40, a useful biomarker in many diseases with a proinflammatory state, has also been found to be elevated in MPN. Moreover, it increases with disease progression and transition to post-PV MF and correlates with neutrophils, platelets, CRP, LDH, and the *JAK2V617F* allele burden [48,73,74]. Neutrophil gelatinase-associated lipocalin (NGAL, also known as lipocalin-2), another useful marker in many inflammatory and malignant diseases (esp. CML), has also been shown to be elevated in patients with PV and ET, and to correlate with neutrophil levels [75,76]. Even higher levels of lipocalin-2 were reported in patients with MF [77]. Experiments by two different groups independently described secretion of NGAL in *JAK2V617F*-harboring cells, leading to a relative increase in proliferation of malignant hematopoietic stem cells compared to non-mutant cells, as the *JAK2V617F* mutation conferred resistance against NGAL-induced apoptosis [77,78]. NGAL also contributes to bone marrow fibrosis by stimulating proliferation of stromal cells [77]. Table 1 summarizes different mediators of inflammation involved in MPN.

Leukocytosis, and especially an increase in leukocytosis, is a useful biomarker to assess risk for thrombosis and bleeding in patients’ MPN [13,79,80]. Table 2 summarizes critical areas of neutrophil involvement. Neutrophilic leukocytosis has been described as marking the transition to high-risk post-PV MF and to be a negative predictive marker for OS [81]. In the same study, increased granulocytic proliferation within the bone marrow was seen. As reviewed by Falanga et al. in 2005, constant activation of neutrophils in PV and ET leading to facilitated endothelial adhesion and aggregation with thrombocytes is thought to contribute to thrombosis [82]. This is evidenced by the increased expression of CD11b/CD18 (a pattern recognition receptor important for adhesion as well as phagocytosis) by neutrophils obtained from patients with ET and PV [83]. In a mouse model, neutrophilia, and especially neutrophil activation, contributed to plaque formation and accelerated atherosclerosis by adhesion and entry into the plaque [84]. Like other inflammatory states, premature atherosclerosis is thought to occur [50]. As a side note, respiratory burst was observed to be impaired in these models, implicating neutrophil dysfunction [83,85]. Evidence obtained from in vitro experiments further suggests an activation of beta-1 integrins by JAK2V617F, implying a role of the mutation in abnormal endothelial–neutrophil interaction [86]. Subsequently, it was shown that venous embolism and splenic sequestration of neutrophils in JAK2V617F-positive mice was prevented by application of beta integrin neutralizing antibodies [87]. Additionally, the activation of platelets by leukocytes, including neutrophils, is thought to contribute to platelet dysfunction and activation and may contribute to a prothrombotic state [88,89]. An increase in platelet numbers in the absence of leukocytosis does not appear to elevate the risk for thrombosis [90]. In the peripheral blood of patients across all MPN subtypes, an elevation in leukocyte or neutrophil-platelet aggregates has been observed as an expression of their interaction [91,92]. Thrombo-inflammation initiated by neutrophils has become a generally accepted mechanism for thrombosis, not limited to MPN [93]. On a bone marrow level, a model has been suggested, whereby megakaryocytic sequestration of neutrophilic (and eosinophilic) granulocytes results in bone marrow fibrosis via activation of fibroblasts, thus contributing to the pathogenesis of PMF [94]. This process is known as emperipolesis and is a physiologic phenomenon, the significance of which is still poorly understood, which is augmented in inflammatory states [95]. In the aforementioned study, abnormal P-selectin expression was described to mediate neutrophil–megakaryocyte interaction, which led to a release of alpha granules and growth factors [94]. Another research group described a significantly elevated percentage of neutrophil-containing megakaryocytes within the bone marrow and spleen in a GATA-1 PMF mouse model [96]. Here, emperipolesis-mediated degranulation as a mechanism for the induction of fibrosis was confirmed, and the authors described an additional mechanism for fibrosis, whereby paraptosis and neutrophil degranulation induce inflammation via TGF-beta [96]. Almost thirty years ago, a high percentage of bone marrow specimens from patients with different MPN subtypes were reported to show emperipolesis: 75% of patients with PV and 100% of patients with ET [97]. As 75% of cases with reactive thrombocytosis also showed this, a correlation between emperipolesis and thrombocytosis seems plausible [97]. In vitro experiments published in 2019 reported the reliance on neutrophil beta-integrin (CD18) expression for emperipolesis [98]. As thrombopoietin alone was unable to increase emperipolesis, additional neutrophilic activation, as there is during inflammation, was hypothesized to be necessary. Emperipolesis also seemed to stimulate thrombopoiesis. Apart from this, the same study described the interesting phenomenon of neutrophils transporting parts of their membrane onto megakaryocytes and onto circulating platelets.

In a mouse model, thrombosis was attributed to increased neutrophilic extracellular trap (NET) formation and prevented by JAK-inhibitor treatment [99]. By the same research group, the importance of JAK2V617F was underlined by a statistically significant association between healthy individuals carrying it as a CHIP and the occurrence of venous thromboembolism. NETs have been studied in many inflammatory processes, including infectious diseases, inflammatory states, and autoimmune disorders [100]. Although helpful for combating bacteria, predominantly harmful features have been attributed to NETs in sterile inflammation, including promoting the occurrence of malignancies [101]. A role in both arterial and venous thromboembolism in different conditions has been ascribed to NETs [102]. Generally, they are formed by neutrophils upon the release of chromatin together with components of their membrane and granules, which subsequently leads to sequestration of erythrocytes and thrombocytes and activation of the coagulation cascade. Despite the potential relevance of NET formation in MPN, the current literature is not entirely clear on its role, as reviewed by Ferrer-Marín et al. [103]. Of the three studies conducted on samples obtained from MPN patients, only one demonstrated increased NET production by neutrophils [99,103,104,105]. An explanation for this discrepancy may lie in the fact that JAK2 inhibitor treatment appears to downregulate NET formation, and several patients of the two negative studies received treatment with JAK2 inhibitors [103].

The fact that neutrophilia in MPN is at least partially reflective of ongoing inflammation is underlined by newer gene expression profiles obtained from individuals with MPN [106]. Although similarities between neutrophilic activation in MPN and G-CSF stimulation exist, an increased activation of inflammatory pathways in neutrophils obtained from MPN patients, in comparison to metabolic pathways in G-CSF mobilized neutrophils from healthy controls, was observed [106].

## 5. Principles of Clinical Management and Conclusions

### 5.1. Phlebotomy and Cytoreductive Therapy

For ET and PV, the prevention of thromboembolic events is a major treatment goal [9]. Evidence supports the use of low-dose aspirin in all patients [107,108]. Aspirin has been shown to decrease the percentage of circulating neutrophil–platelet complexes, thereby reducing the reciprocal activation of these two cell types [91]. In PV, phlebotomy with a hematocrit goal below 45% significantly reduces the risk for cardiovascular events [109]. Cytoreductive therapy (e.g., hydroxyurea) should be used in patients at higher risk [9]. These agents result in a decline of all cell lines, including neutrophilic granulocytes, and might thereby indirectly reduce inflammation. Hydroxyurea may also be used in low-risk PV or ET for aspirin-refractory patients [9]. Additionally, it may be used as treatment of splenomegaly in MF [47]. Moreover, according to a study from 2010, 82% of patients will experience a relief of constitutional symptoms [110]. Pancytopenia, however, may be aggravated by the use of hydroxyurea. A selective reduction of platelets in patients with ET can be achieved with anagrelide, which has been compared to hydroxyurea in ET [111,112]. One randomized trial reported an increase in arterial but a decrease in venous thromboembolic events, and an increased risk for fibrotic transformation for anagrelide compared to hydroxyurea [111]. This implies the involvement of other cell lines in the development of such complications, possibly leukocytes. A more recent clinical trial, however, did not show significant differences in thrombo-hemorrhagic events between patients treated with anagrelide versus hydroxyurea and met the primary endpoint of anagrelide’s non-inferiority [112].

### 5.2. Interferon

Interferon alpha is another cytoreductive agent which has been shown to improve constitutional symptoms and blood counts, as well as to decrease *JAK2* allelic burden [113,114]. Although its exact mechanism of action is still subject to research, interferon alpha is thought to preferentially induce cell death in malignant progenitors and has been believed to lead to a depletion of malignant stem cells, as well as to exert an immunomodulatory effect on several inflammatory cells [115]. Moreover, ropeginterferon alfa-2b (R-2b) is approved independently of prior treatment with hydroxyurea [116]. The PROUD-PV and CONTINUATION-PV phase III clinical trials reported the complete hematologic response and the molecular remission to be higher for R-2b than for hydroxyurea after 36 months of treatment [117]. In another clinical trial, significantly improved control of hematocrit with addition of R-2b to phlebotomy was observed [118].

### 5.3. JAK Inhibitors

Therapeutic approaches to PV (in case of failure of other available therapies) and PMF include JAK2 inhibitor treatment, most notably ruxolitinib, and, more recently, fedratinib [119]. Importantly, they are effective irrespectively of the presence of a JAK2V617F mutation. A significant reduction in constitutional symptoms was reported by the landmark clinical trials for ruxolitinib in PMF and PV, as well as a significant improvement of splenomegaly and prolongation of OS in PMF and control of hematocrit in PV [120,121,122,123]. In the COMFORT-I trial, 41.9% of MF patients treated with ruxolitinib experienced a reduction in spleen volume of at least 35% (versus 0.7% in the placebo group) and a 45.9% reduction in general symptoms (versus 5.3%) [120]. Similarly, the COMFORT-II trial reported a spleen volume reduction of at least 35% in 28% of patients, compared to 0% of patients receiving best available treatment, and the responses were ongoing during a long-term follow up [123,124]. For PV, in the RESPONSE trial, 21% of ruxolitnib-treated patients reached the primary end point (control of hematocrit and a spleen volume reduction of 35%), compared to only 1% of patients with the best available treatment. In the RESPONSE-II trial, hematocrit control was achieved in 62% with ruxolitinib versus 19% with the best available treatment [121,122]. Currently, the role of JAK inhibitors in the treatment of PV is minor compared to its role in treating MF.

Fedratinib has been approved in ruxolitinib-resistant MF after the phase II JAKARTA-2 trial had reported a decrease in spleen volume by one third in 55% of ruxolitinib-pretreated patients [125]. Additionally, a significant improval of symptoms was described [126]. Whereas fedratinib is a very selective JAK2 inhibitor, ruxolitinib inhibits JAK1 and JAK2 [127].

Two novel JAK inhibitors are currently under investigation: Pancratinib (a JAK2 and FLT3 inhibitor) is effective in the reduction in spleen volume and symptoms in MF, and is especially promising in the treatment of patients with baseline cytopenias, which is usually a major limitation for treatment [128]. It has been shown to be more effective in comparison to the best available treatment, including ruxolitinib, and was also effective in cases of a low *JAK2V617F* allele burden [129,130].

Momelotinib, apart from being an JAK1/2 inhibitor, antagonizes the effects of hepcidin; therefore, it is thought to become useful in MF with transfusion-requiring anemia. Although it decreased transfusion dependency and was non-inferior to ruxolitinib regarding spleen volume reduction, greater symptom control was achieved with ruxolitinib in a phase III trial [131]. For ruxolitinib-pretreated patients, momelotinib significantly decreased transfusion dependency and symptoms, but did not decrease spleen volume when compared to the best available therapy [132].

Although JAK2 inhibitors provide alleviation of symptoms, their ability to induce complete hematologic remission is limited, and complete molecular remission is not observed [133]. Additionally, disease progression (i.e., transformation into MF and AML) can occur despite therapy. As reviewed by Greenfield and colleagues, an explaination for the clinical benefit of ruxolitinib may lie in the reduction in ongoing inflammation rather than selective inhibition of the disease’s driver mutation, and it has even been described as lacking antitumor activity [47,133]. Unsurprisingly, JAK inhibitors have therefore proven to be effective in a variety of other (auto-)inflammatory conditions [47]. These include hemophagocytic lymphohistiocytosis, which is characterized by overproduction of proinflammatory cytokines and hyperactivation of macrophages, as well as graft-versus-host disease (GvHD) [134]. JAK signaling has been demonstrated to be important for T-cell activation and proliferation in GvHD via the production of proinflammatory cytokines and activation of neutrophilic granulocytes [135]. Clinical trials have reported encouraging results for ruxolitinib in steroid-refractory patients with GvHD [135,136].

The clinical presentation of patients with MPN depends, apart from the stage of disease and comorbidities, on the underlying molecular biology of the disease. We have discussed how mutations in *JAK2*, *MPL*, and *CALR* influence the phenotype of disease and how additional mutations influence the course of disease. An increase in leukocytosis, largely determined by the levels of neutrophils, is a risk factor for the development of thrombotic and hemorrhagic complications [13,79,80]. Mechanisms have been described, according to which neutrophils cause thrombotic events via NET formation and via interactions with endothelial cells and thrombocytes [82,86,99]. Chronic inflammation is understood to impact the initiation and progression of disease in a major way, and neutrophils appear to contribute to bone marrow fibrosis [96]. A further dissection of the complex interactions that neutrophils are involved in is necessary in order to establish novel treatment strategies as the currently available options do not specifically influence these processes.

## Figures and Tables

**Figure 1 ijms-22-09555-f001:**
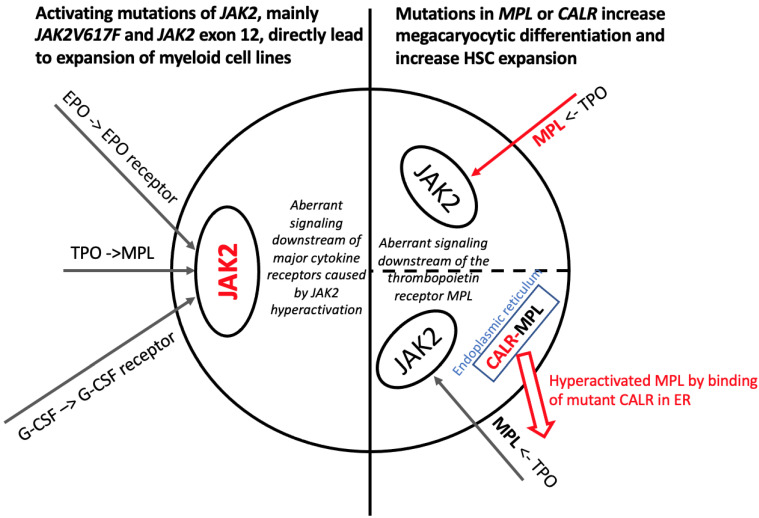
Schematic overview of hallmark mutations in MPN. JAK2 lies downstream of the EPO receptor, the thrombopoietin receptor (MPL), and the G-CSF receptor. The activation of JAK2 by a JAK2V617F or a JAK2 exon 12 mutations, therefore, enhances signaling downstream of pathways that would normally be activated by the growth factors for erythropoiesis, thrombopoiesis, or granulopoiesis. An activating mutation in MPL leads to increased signaling through the thrombopoietin (TPO)-thrombopoietin receptor-(MPL)-axis. Similarly, a mutation in the chaperone protein CALR leads to a constitutive activation of MPL upon binding of CALR to MPL in the endoplasmic reticulum (ER).

**Table 1 ijms-22-09555-t001:** Overview of the roles of important inflammatory markers in MPN.

Inflammatory Cell/Organelle/Protein	Role
Bone marrow stromal cells [52]	Expansion of *JAK2V617F* mutant HSC
Inflammasome [53]	Inflammation aiding disease progression
TNF-alpha [54]	Expansion of *JAK2V617F* mutant HSC and suppression of non-mutant HSC
IL-33 [55]	Prolonged survival of *JAK2V617F*-positive cell lines
IL-11, HGF [56]	Necessary for expansion of erythroid precursors harboring *JAK2V617F*
IL-6, CXCL10, FGF [57]	Protection of malignant clones against JAK2 inhibition
NF-E2 [60,63]	Leuko- and thrombocytosis, increases IL-8
IL-8 [65,66,67] 0.75	Alteration of megakaryopoiesis, neutrophilia, erythrocytosis
CRP [70,71,72]	Correlation with *JAK2V617F* allele burden in PV, ET, PMFPrognostic marker in PMF
IL-8, IL-2R, IL-12, IL-15 [69]	Prognostic markers in PMF
YKL-40 [48,73,74]	Correlated with disease progression, transition to post-PV MF, neutrophilia, thrombocytosis, CRP, LDH, *JAK2V617* allele burden
Neutrophil gelatinase-associated lipocalin (NGAL) [75,76,77]	Correlation with neutrophilia in PV, ET, MF

**Table 2 ijms-22-09555-t002:** Roles of neutrophilic granulocytes and leukocytes in MPN.

Major Mechanism	Area of Involvement
Increased endothelial adhesion and accelerated atherosclerosis [82,86]	Arterial thrombosis
Increased NET formation [99]	Thrombosis
Aggregation of thromboyctes with neutrophils [82]	Thrombosis
Emperipolesis [94,96]	Bone marrow fibrosis, thrombocytosis

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
