# Peer review of "The Role of Neutrophilic Granulocytes in Philadelphia Chromosome Negative Myeloproliferative Neoplasms"

_ijms, 2021, doi:10.3390/ijms22179555_

Round 1

Reviewer 1 Report

The review „The role of neutrophilic granulocytes in Philadelphia chromosome negative myeloproliferative neoplasms“ of Kiem and colleagues aims to summarize the current understanding of the role of leukocytosis and especially neutrophils in MPN. In general, a review on this topic is highly interesting.  The manuscript includes a huge summary of citations but in most cases the authors insufficiently discuss these studies.

And that’s my biggest concern: The title is not fitting to the content of the manuscript. Instead of having an in depth discussion of leukocytosis and neutrophils in MPN, the authors try to include too many aspects: secondary mutations, clonal hematopoiesis, treatment. All these topics are important but should be discussed in relation to granulocytes. This was partly done but instead of going into detail, the next topics were covered. In addition, none of the figure(s) or tables are neutrophil specific. With this title and abstract, I would expect at least one figure summarizing the current data on leukocytosis/neutrophils in MPN. Therefore, the manuscript should be adjusted in a more detailed way, topicwise. In addition, in the second part, language/wording needs to be improved.

Here are some observations:

  1. Lane 28: CEL-NOS or CEL, NOS
  2. Gene names should be in italics
  3. Lane 29: …(CEL-NOS) and MPN unclassifiable (MPN-U).
  4. At several positions in the manuscript, the authors are mentioning important data on granulocytes but writing in parentheses. Why?
  5. Lanes 55-56: …we will elucidate…
  6. Chapter 3 on neutrophilic granulocytes is not detailed at all or should be included into other chapters.
  7. Lanes 80-81: aforementioned genes. (not pathways)
  8. Lane 90: lineages instead of lines?!
  9. Lane 116: …JAK2V617F as the initial…
  10. Lane 142: Please include abbreviation: “overall survival (OS)”
  11. Figures and tables are not mentioned in the text. And in the figure legend of figure 1 are information included not presented in the figure itself.
  12. Lane 220….: This is called emperipolesis. Why not discussing this phenomenon in more detail, as granulocytes are involved and several papers exist on this topic.
  13. Lanes 225-228: not meaningful in this brevity
  14. Lane 234: neutrophilic extracellular trap formation. Again, pretty interesting but only marginally discussed
  15. Table 1: The role of LCN2 (NGAL) was also discussed in MF (Lu M et al, Blood 2015)
  16. Topic 7: Please bring treatment into the context of granulocytes/leukocytosis.
  17. Check lanes 250-252, 313, 324, 330-331 (hematologic response; molecular response), 352-354, 365-366 374 for language.
  18. Lane 324: improvement
  19. Lane 378: TNFR2 is no cytokine! Or is the wording just confusing?
  20. Lanes 370-398 is not well written.
  21. Abbreviations: lane 410 lymphoma; lane 420 essential thrombocythemia

Author Response

Reply

We thank the reviewer for his critical feedback. In the course of this revision, we have made adjustments to focus our attention more towards the involvement of neutrophilic granulocytes in MPN. We extended the chapter on “Inflammation and Neutrophils in MPN” and significantly reduced the discussion on therapy. A new table on the roles of neutrophils has been created and the table on therapeutic options has been deleted. 

  1. Lane 28: CEL-NOS or CEL, NOS

This has been corrected (line 29).

  1. Gene names should be in italics

The according adjustments in the entire document have been made.

  1. Lane 29: …(CEL-NOS) and MPN unclassifiable (MPN-U).

We have inserted these abbreviations. (line 29)

  1. At several positions in the manuscript, the authors are mentioning important data on granulocytes but writing in parentheses. Why?

We thank the reviewer for this interesting observation and have changed this as appropriate. This has substantially improved comprehensibility.

  1. Lanes 55-56: …we will elucidate…

This has been corrected.

  1. Chapter 3 on neutrophilic granulocytes is not detailed at all or should be included into other chapters.

We agree with the reviewer and the entire Chapter 3 has been incorporated into the chapter on “Molecular biology of Ph-negative myeloproliferative neoplasms and predictive/prognostic implications”

  1. Lanes 80-81: aforementioned genes. (not pathways)

This has been corrected.

  1. Lane 90: lineages instead of lines?!

This has been corrected.

  1. Lane 116: …JAK2V617F as the initial…

This has been corrected.

  1. Lane 142: Please include abbreviation: “overall survival (OS)”

The abbreviation has been included.

  1. Figures and tables are not mentioned in the text. And in the figure legend of figure 1 are information included not presented in the figure itself.

We appreciate this comment, and all figures are now briefly mentioned in the main text. Also, the figure legend for figure 1 has been reduced to optimally describe the figure.

  1. Lane 220….: This is called emperipolesis. Why not discussing this phenomenon in more detail, as granulocytes are involved and several papers exist on this topic.

We thank the reviewer for this suggestion and have included a paragraph discussing the role and mechanism of emperipolesis in MPN. In fact, we regard the deeper discussion of emperipolesis to significantly contribute to the improvement of our paper.

(line 362-375: “This process is known as emperipolesis and is a physiologic phenomenon of still poorly understood signifcance, which is augmented in inflammatory states.102 In the aferometioned study, abnormal P-selectin expression was described to mediate NG-megakaryocyte interaction, which led to release of alpha granules and growth factors.101 Another group described significantly elevated percentage of NG-containing megakaryocytes within bone marrow and spleen in a GATA-1 PMF mouse model.103 Here, emperipolesis-mediated degranulation as mechanism for induction of fibrosis was confirmed and authors described an additional mechanism for fibrosis, whereby paraptosis and NG degranulation induce inflammation via TGF-beta.103 Almost thirty years ago already, a high percentage of bone marrow specimens from patients with different MPN subtypes MPN were reported to show emperipolesis: 75 % of patients with PV and 100% of patients with ET. As 75% of cases with reactive thrombocytosis did as well, a correlation of emperipolesis and thrombocytosis seems plausible.104 In vitro experiments published in 2019 reported the reliance on NG beta-integrin (CD18) expression for emperipolesis.105 As thrombopoietin alone was unable to increase emperipolesis, additional neutrophilic activation, as ther is during inflammation, was hypothesized to be a necessity. Apart from that, the same study described the interesting phenomenon of NG transporting parts of their membrane onto megakaryocytes as well as onto circulating platelets. Also, emperipolesis seemed to stimulate thrombopoiesis.”)

  1. Lanes 225-228: not meaningful in this brevity

This is a valid observation, which led us to extend the discussion on the difference in neutrophil activation in MPN compared to growth factor stimulation.

(line 414-419:“The fact that neutrophilia in MPN is at least partially reflective of ongoing inflammation is underlined by newer gene expression profiles obtained from individuals with MPN. Although similarities between neutrophilic activation in MPN and with G-CSF stimulation exist, an increased activation of inflammatory pathways in NG obtained from MPN patients in comparison to metabolic pathways in G-CSF mobilized NG from healthy controls was observed Notable alterations included altered in expression of different proteins within NF-kappaB signaling pathway, with upregulation of some but downregulation of others, making it difficult to draw a conclucsion regarding the role of this pathway. 99

  1. Lane 234: neutrophilic extracellular trap formation. Again, pretty interesting but only marginally discussed.

This is very valuable input, which is why we have included a much extensive discussion on NET formation, as NETs appear to play a key role for thrombotic events in MPN. (line 380-413): “NETs have been studied in many inflammatory processes, including infectious disesaes, inflammatory states and autoimmune disorders.103 Although helpful for combating bacteria, predominantely harmful features have been attributed to NETs in sterile inflammation, including supporting malignancies.104 A role in both arterial and venous thrombembolism in different conditions has been described for NETs.105 Generally, they are formed by NG upon release of chromatin together with components of their membrane and granules, which subsequnently leads to sequestration of erythrocytes and thromobcytes and activation fo the coagulation cascade. Despite a potential relevance of NET formation in MPN, current literature is not entirely clear on its role, as reviewed by Ferrer-Marín et al..106 Of the three studies conducted on samples obtained from MPN patients, only the one cited above demonstrated increased NET production by NG. 102, 106-108 An explanation for this discrepency may lie in the fact that JAK2 inhibitor treatment appears to downregulate NET formation, but several patients of the two negative studies received JAK2 inhibitors.106

  1. Table 1: The role of LCN2 (NGAL) was also discussed in MF (Lu M et al, Blood 2015)

We appreciate the suggestion of this interesting paper and have included the suggested paper and added further information to our article:

Line 321-327: “Even higher levels of lipocalin were reported in patients with MF.84 Experiments by two different groups independently described secretion of NGAL in JAK2V617F-harboring cells, leading to a relative increase in proliferation of malignant hematopoietic stem cells over non-mutant cells, as the JAK2V617F mutation seemed to confer resistance against NGAL-induced apoptosis.84, 85 NGAL also contributes to bone marrow fibrosis by stimulating proliferation of stromal cells.84 Table 1 summarizes different mediators of inflammation involved in MPN.

  1. Topic 7: Please bring treatment into the context of granulocytes/leukocytosis.
  2. Check lanes 250-252, 313, 324, 330-331 (hematologic response; molecular response), 352-354, 365-366 374 for language.
  3. Lane 324: improvement
  4. Lane 378: TNFR2 is no cytokine! Or is the wording just confusing?
  5. Lanes 370-398 is not well written.

16.-20. We understand that the discussion on therapeutic options is too extensive. We have deleted large parts of this section and only included information with more direct connection to neutrophilic inflammation and hope that this will aid in making our paper more concise. We once again thank the reviewer for this important contribution!

  1. Abbreviations: lane 410 lymphoma; lane 420 essential thrombocythemia

These abbrevations have been included

Reviewer 2 Report

Unfortunately the abstracts does not fit the content of the review. I do not agree that the role of neutrophilic granulocytes in MPN is described and discussed well in the paper – one half of the paper is focused only on clinical managements, conclusion is rather vague. Section 5 is properly focused and brings review-like interesting information, the other sections are just mixture of common information. There are too many spelling and typing mistakes, the language is often inappropriate (52, important complications …, 55 will or would, 89 how lymphoid cells can underline pluripotency of HSCs?, V617F in?, 119 rephrase generally be acquired later, 127 associated, 142 OS only abbreviation, 260 years years, 268 hydroxyurea, elevated, 271 patients patients, 292 significantly, 297 ruxolitinib, 302 myelofibrosis …

The topic is interesting, section 5 is great, section 1-4 are acceptable, section 6 is missing :) and section 7 needs to be rewritten or replaced by more deep, accurate information, e.g. how the neutrophilic granulocytes are affected, treated, or reacts on clinical management.

Author Response

We thank the reviewer for making many important suggestions. The abstract has been rewritten to better fit the article. We have eliminated large parts of the discussion on therapy and slightly reduced the discussion on the underlying molecular biology. We extended the discussion on molecular and cellular events neutrophilic granulocytes are involved, especially by bringing more attention to emperipolesis and to neutrophilic extracellular tap formation. Also, the table on therapeutic options was deleted and a new table summarizing key events involving neutrophilic granulocytes was added.

In addition to that, we are grateful for the observations made regarding language and have corrected the mistakes. We believe that due to the substantial changes made, our paper has significantly improved and brings much more attention to the main topic, i.e. neutrophilic granulocytes.

Round 2

Reviewer 1 Report

I am sorry, but the manuscript has not been improved but is more patchwork now. Only the second part has been improved, but still needs to be streamlined. The whole manuscript is full with typos and a native-speaker needs to correct the English language. I cannot accept it at all in the current version.

Comments to new abstract:

The authors transferred one sentence and included another, and the connection of the sentences is sparse. Why has the last sentence been deleted? The abstract needs to be optimized.

  1. Lane 13: include abbreviation: MPN
  2. Lane 16: „They“ needs to be changed to “MPNs” due to the missing connection to the novel sentence in lanes 14-16.
  3. Revise new sentence in lanes 21-23.

Further comments (by far not all):

Topic 2.

Lanes 85-63: Please correct the poor English. Is nobody reading the final manuscript before submission?!

Lane 62: I’ve never seen the abbreviation NG for neutrophilic granulocytes. Why notjust using neutrophils?

Topic 3.

Lanes 76-84: English language needs to be optimized! But most of all, the common theme is lost. The former topic 3 was not integrated into the new topic 3, it was only copy/pasted.

Lane 78: use abbreviation à MPN

Lane 79: The differentiation block takes place in the malignant cells. Why using parentheses here?

Lanes 102-103: Language

Lanes 120-123: Language

Figure 1: The legend should explain what is shown in the Figure. ER, G-CSF and EPO receptor, exon 12 mutations, TPO…Please explain concisely.

Topic 4

The topic is on inflammation and neutrophilic granulocytes (see title) and still parentheses were used for neutrophils in lines 163-164. Why? This is your main topic and the title of the manuscript.

Lanes 176-178: language

Lane 179: IL1beta cannot promote primary myelofibrosis (PMF) in vitro. It may induce myelofibrosis, fibrotic tissue formation or similar…Please rephrase.

Lane 196: lipocalin-2. Lipocalins describe a protein family.  

Lane 199: …conferred resistance….

Table 1: tables needs to have a header.

Author Response

We thank the reviewer his time and efforts to provide us with detailed feedback and helpful comments. These have resulted in several major changes in our manuscirpt. Importantly, we have reduced the discussion on the molecular biology underlying MPN that had no immediate relationship with neutrophils. Also, several parts of the manuscript have been rewritten to improve language. We believe that the manuscript has therefore critically improved in quality, structure and readability.

Point-by-point discussion of the reviewer’s comments:

Abstract

  1. The authors transferred one sentence and included another, and the connection of the sentences is sparse. Why has the last sentence been deleted? The abstract needs to be optimized.

1.Lane 13: include abbreviation: MPN

2.Lane 16: „They“ needs to be changed to “MPNs” due to the missing connection to the novel sentence in lanes 14-16.

3.Revise new sentence in lanes 21-23.

We thank the reviewer for his feedback regarding the abstract. The missing abbreviation has been included and the second half of the abstract has been rewritten to improve coherence between sentences. Also, we have re-inserted the prior last sentence that pointed out the goal of the review. “Chronic inflammation impacting initiation and expansion of disease in a major way has been described. Neutrophilic granulocytes play a major role in the pathogenesis of thromboembolic events via secretion of inflammatory markers as well as via interaction with thrombocytes and the endothelium. In this review, we discuss the molecular biology underlying myeloproliferative neoplasms and point out the central role of leukocytosis and, specifically, neutrophilic granulocytes in this group of disorders. (lane 18-23)”

Topic 2

  1. Lanes 85-63: Please correct the poor English. Is nobody reading the final manuscript before submission?!

We apologize for the poor language and we have rewritten this paragraph.

Lane 62: I’ve never seen the abbreviation NG for neutrophilic granulocytes. Why not just using neutrophils?

We thank the reviewer for this important comment. We have reacted to this by changing NG to the more common term neutrophils in the entire document.

Topic 3.

Lanes 76-84: English language needs to be optimized! But most of all, the common theme is lost. The former topic 3 was not integrated into the new topic 3, it was only copy/pasted.

Lane 78: use abbreviation à MPN

Lane 79: The differentiation block takes place in the malignant cells. Why using parentheses here?

We thank the reviewer for critically looking into this paragraph. Therefore, we have now rewritten it and drastically decreased its length. We understand how a discussion on neutrophil differentiation limited to the context of MPN adds quality to our review and hope that this issue has been adequately addressed. Also, the discussion on the molecular biology underlying MPNS with description of several involved genes has decreased in length to not become overwhelming compared to the discussion on neutrophils. The abbreviation has been inserted as recommended. The changes are highlighted within the document

Lanes 102-103: Language

Lanes 120-123: Language

These lanes have been edited.

Figure 1: The legend should explain what is shown in the Figure. ER, G-CSF and EPO receptor, exon 12 mutations, TPO…Please explain concisely.

We regard this to be a very valuable comment and have adapted the legend accordingly. This should add additional clarity to the figure. Also, we eliminated a typo from the figure (thrombopoietin).

(“Figure 1. Schematic overview of hallmark mutations in MPN. JAK2 lies downstream of the EPO receptor, the thrombopoietin receptor (MPL) and the G-CSF receptor. The activation of JAK2 by a JAK2V617F or a JAK2 exon 12 mutations therefore enhances signaling downstream of pathways that would normally be activated by the growth factors for erythropoiesis, thrombopoiesis or granulopoiesis. An activating mutation in MPL leads to increased signaling through the thrombopoietin (TPO)-thrombopoietin receptor-(MPL)-axis. Similarly, a mutation in the chaperone protein CALR leads to a constitutive activation of MPL upon binding of CALR to MPL in the endoplasmic reticulum (ER).”)

Topic 4

The topic is on inflammation and neutrophilic granulocytes (see title) and still parentheses were used for neutrophils in lines 163-164. Why? This is your main topic and the title of the manuscript.

We apologize for this inaccuracy. In line with the rest of the document, the term neutrophil has been inserted.

Lanes 176-178: language

We have rewritten this part to provide a clearer description of the process: “The transcription factor NF-E2 is often found to be overexpressed in patients with MPN63, independently of JAK2 mutational status. As a mechanism, an increased binding of RUNX-1 to the NF-E2 promotor has been described in neutrophils.64

Lane 179: IL1beta cannot promote primary myelofibrosis (PMF) in vitro. It may induce myelofibrosis, fibrotic tissue formation or similar…Please rephrase.

As kindly suggested, the more adequate term “induce” is now used. “NF-E2is also induced by IL-1beta, mainly found in PMF and ET and shown to induce myelofibrosis and fibrotic tissue formation in vitro66, as well as to be secreted by JAK2V617F-positive HSC in animal models67 (lane 655-657).

Lane 196: lipocalin-2. Lipocalins describe a protein family. 

We thank the reviewer for his meticulous review and have eliminated this inaccuracy by inserting the correct term as recommended (lane 677)

Lane 199: …conferred resistance….

This was corrected (lane 680).

Table 1: tables needs to have a header.

We appreciate this observation and have attributed headers to both tables

Round 3

Reviewer 1 Report

The manuscript was decisively improved. Thank you for implementing my thoughts and comments. 

Here are only a few final comments:

Please also include the abbreviations for polycythemia vera and essential thrombocythemia in the abstract.

Lane 47: …, and (or which!?) will be described….

Lane 48: Section 3.

Lanes 67-68: Probably you can transfer this sentence to the end of section 3.

Lane 111: Delete one “with”.

Lane 242 and 262: Citations are missing

Lane 263:- delete “with”

Lanes 261-267: The paragraph is rather short. You may delete the NFkB part or discuss in more detail, probably in comparison to the JAK/STAT pathway.

Lane 284: better use “cell types” than “cell lines”

Lanes 299 and 302: interferon-alpha and interferon alpha

Lane 315 and 339: Use abbreviation for myelofibrosis

Lane 274-279: I would suggest to combine this part with lanes 349-353 to have a coherent paragraph for the conclusion.

Author Response

We thank the reviewer for his positive feedback. We believe that the implementation of the reviewer’s additional kind suggestions further improves our manuscript. Few changes in the article’s structure were made as proposed, which will enhance readability and clarity. Moreover, formal inaccuracies (abbreviations, citations etc.) were eliminated. A detailed point-by-point description can be found below.

Please also include the abbreviations for polycythemia vera and essential thrombocythemia in the abstract.

We are happy to include these abbreviations into lane 14.

Lane 47: …, and (or which!?) will be described…. & Lane 48: Section 3.

We thank the reviewer for this suggestion, we have changed this into “...other genetic factors will be described in more depth in section 3.“ (lane 51-52)

Lanes 67-68: Probably you can transfer this sentence to the end of section 3.

This is an interesting observation; we have gladly transfered the sentence to lanes 104-106.

Lane 111: Delete one “with”.

This has been corrected.

Lane 242 and 262: Citations are missing

We are grateful that our attention has been drawn towards this and we have inserted the missing citations. In both cases, the citation had erroneously been attached to the following sentence only.

Lane 263:- delete “with”

This has been corrected.

Lanes 261-267: The paragraph is rather short. You may delete the NFkB part or discuss in more detail, probably in comparison to the JAK/STAT pathway.

We thank the reviewer for this important comment. As this paragraph represents a side note only, we have decided to simplify it further and have deleted the last part mentioning NFkB.

Lane 284: better use “cell types” than “cell lines”

We see how this is a better expression and are happy to change the wording to “cell types”.

Lanes 299 and 302: interferon-alpha and interferon alpha

This is an inaccuracy on our part; we have reacted to this by changing to “interferon alpha” in lane 299.

Lane 315 and 339: Use abbreviation for myelofibrosis

We thank the reviewer for this observation and have inserted the abbreviation MF.

Lane 274-279: I would suggest to combine this part with lanes 349-353 to have a coherent paragraph for the conclusion.

We thank the reviewer for this interesting suggestion. The two parts have been combined with slight adaptations. This improved our conclusion and will draw a clearer picture of the described processes. “The clinical presentation of patients with MPN depends, apart from stage of disease and comorbities, on the underlying molecular biology of the disease. We have discussed how mutations in JAK2, MPL and CALR influence the phenotype of disease and how additional mutations influence the course of disease. An increase in leukocytosis, largely determined by the levels of neutrophils, is a risk factor for the development of thrombotic and hemorrhagic complications. Mechanisms have been described, according to which neutrophils cause thrombotic events via NET formation and via interactions with endothelial cells and thrombocytes. Chronic inflammation is understood to impact initiation and progression of disease in a major way and neutrophils appear to contribute to bone marrow fibrosis. A further dissection of the complex interactions that neutrophils are involved in is necessary in order to establish novel treatment strategies, as currently available options do not specifically influence these processes.” (lane 306-314)